# The Role of Preschool Hours in Achieving Physical Activity Recommendations for Preschoolers

**DOI:** 10.3390/children8020082

**Published:** 2021-01-25

**Authors:** Sara Lahuerta-Contell, Javier Molina-García, Ana Queralt, Vladimir E. Martínez-Bello

**Affiliations:** 1Conselleria d’Educació, Generalitat Valenciana, 46015 Valencia, Spain; saconla@alumni.uv.es; 2COS Research Group, Body, Movement, Music and Curricular Practices, University of Valencia, 46022 Valencia, Spain; 3Department of Teaching of Musical, Visual and Corporal Expression, University of Valencia, Avda. dels Tarongers, 4, 46022 Valencia, Spain; Javier.molina@uv.es; 4AFIPS Research Group, University of Valencia, 46022 Valencia, Spain; ana.queralt@uv.es; 5Department of Nursing, University of Valencia, Jaume Roig, s/n, 46010 Valencia, Spain

**Keywords:** preschool, physical activity, school hours, educational settings, structured movement sessions, early childhood education

## Abstract

Research on physical activity (PA) in different educational settings could elucidate which interventions promote a healthy school lifestyle in early childhood education (ECE). The aims of this study were: (a) to analyse the PA levels of preschoolers during school hours, as well as the rate of compliance with specific recommendations on total PA (TPA) and moderate-vigorous PA (MVPA); (b) to examine the role of structured movement sessions and recess time in the MVPA levels during school hours; (c) to evaluate the sociodemographic correlates of preschoolers and the school environment on MVPA behaviour during school hours. PA was evaluated with Actigraph accelerometers. Our main findings were that: (a) preschoolers engaged in very little TPA and MVPA during school hours; (b) children showed significantly higher MVPA levels on days with versus without structured movement sessions, and the contribution of the structured sessions to MVPA was significantly higher than that of recess time; (c) gender and age were associated with PA, and a high density of young children on the playground was associated with high levels of vigorous PA, whereas in the classroom, high density was associated with more sedentary behaviour. Structured PA could reduce the gap in achieving international recommendations.

## 1. Introduction

Physical inactivity and sedentary behaviour (SB) are health risk factors in children [1] and are associated with a higher incidence of obesity, poorer mental health, and a greater likelihood of physical inactivity in adulthood [2,3]. Children aged three to five years need at least 180 min of daily physical activity (PA); at least 60 min of this should be moderate to vigorous PA (MVPA) [4]. Early childhood education (ECE) environments have great potential to promote PA and reduce SB in young children [5], who can potentially get a significant amount of their daily PA in these settings [6,7,8]. However, studies show that the achievement of this target is uneven [9,10,11].

Accelerometer-based studies in preschoolers have examined PA by measuring hour-by-hour patterns [12,13,14,15]; average daily PA levels during specific time periods of the day, such as recess time [16,17]; average daily PA levels on weekdays and weekends [18,19,20]. Furthermore, different PA correlates have been identified in young children [5,21], including lower child-staff ratios and the use of indoor play areas for motor activity [22], and child–staff ratios and the size of indoor area per child [23]. In the ECE context, research indicates that planned PA breaks contribute significantly to time spent in MVPA during the school day [24]. Furthermore, curricular practices in the ECE context (such as the structured time for movement practices) may play an important role in developing motor skills [25]. However, structured movement sessions have not been studied as a discrete segment during the ECE school day to determine their contribution to total PA during school hours and to the achievement of PA recommendations.

School PA is a significant component of daily PA. Because correlates of children’s PA in ECE are multidimensional, research on PA in different educational settings could elucidate which interventions effectively promote a healthy school lifestyle in the ECE context. Some studies have described accelerometer-assessed PA patterns during segmented school weekdays [12,13,14,15,18,19,20,26]. However, there is no evidence on the impact of structured or unstructured activities on total PA (TPA; including not only MVPA but also light PA) or MVPA during the school day. To date, no studies have explained the role of ECE institutions in achieving PA recommendations by differentiating the contributions of structured movement sessions and recess time to total PA levels during school hours. As Andersen et al. (2017) argues, a holistic description of PA patterns using the full range of the accelerometer may help identify intervention opportunities and potentially highlight critical windows through which to intervene in the ECE context [19].

Gender and age have been shown to predict MVPA in preschoolers. The literature indicates that boys are more physically active than girls during the school day [8,11,23], and this pattern also holds true in specific time bands, such as recess [17,27,28] and structured PA sessions [27,28,29]. At the same time, recently Nielsen et al. (2019) found that the effects of the preschool arena on MVPA are more important as children grow up, with older children benefiting more from this environment in terms of high-intensity activities than younger ones [7]. Further research is necessary to empirically examine how to reduce the gender gap in young children in terms of PA promotion and to explain how PA is affected by the relationship between age and the ECE environment.

To our knowledge, there are no studies in our country that objectively measure and describe PA levels in young children attending ECE institutions. Therefore, the aims of this study were threefold: (a) to analyse the PA levels of preschool children during school hours, as well as the rate of compliance with specific recommendations on TPA and MVPA; (b) to examine the role of structured movement sessions and recess time in the MVPA levels during school hours, as well as the adherence to specific recommendations on MVPA; (c) to evaluate the sociodemographic correlates of young children and the school environment on MVPA behaviour in preschool children during school hours.

## 2. Materials and Methods

### 2.1. Design and Participants

The study used a cross-sectional, correlational design. We included 116 young children aged three to four years (mean age 4.3 (SD = 0.5); 59 girls) from 6 different public ECE institutions in the province of Valencia, Spain (Table 1). Inclusion criteria were children aged 3–5 years and being able to walk without assistance. The parents or guardians of all children provided informed consent prior to study commencement. The Human Research Ethics Committee at the corresponding author’s university approved the study protocol (ethical approval code- H1424684423064). Data were collected from January 2017 to April 2019.

### 2.2. Objective Physical Activity Measurement

PA levels were measured with Actigraph accelerometers (GT3X+Actigraph, Pensacola, FL, USA). Accelerometers are an objective measurement device and have been validated to assess PA in the preschool population [8,30]. Children wore the accelerometers on the hip during throughout the school day on 5 consecutive days, including four days without and one day with a movement session. Practitioners noted any comments regarding their use on a study registration sheet. Children (*n* = 20) who did not have 5 consecutive days registered were excluded (e.g., due to illness or lack of attendance at the school on the days when measurements were carried out) from the final analyses. Accelerometry measured PA intensities in 15-s increments, and PA was scored using Van Cauwenberghe’s cut-off points as follows [30]: SB, 0–372 counts per 15 s; light PA, 373–584 counts; moderate PA, 585–880 counts; vigorous PA, ≥881 counts.

### 2.3. School Hours, Structured Movement Sessions and Recess Time

PA was evaluated throughout the school day. The ECE institutions had a schedule from 9:00 am to 2:00 pm. The teachers responsible for each classroom were in charge of placing the accelerometer on the children at the beginning of the school day (9:15 am) and removing it a few minutes before the end (1:45 pm). In addition, we have delimited the measurement of PA in two specific time periods: recess and structured movement sessions. These represent the two specific opportunities for children to be physically active during the school day.

The structured movement sessions were structured (i.e., following guidelines to achieve educational goals) and were part of the curriculum of the preschools participating in the research, that is, they had a justification, objectives, procedure, and evaluation. The teachers structured the sessions according to the guidelines of their own curriculum. The sessions took place once a week, were carried out during the regularly scheduled time period, were held in an indoor classroom, lasted approximately 45 min, and 14 to 20 children participated. The same number of children participated in the session as during the whole school day. On the day of the measurement, the teachers were instructed to conduct the session the way they usually do. The indoor classroom was equipped with typical early childcare equipment in order to stimulate children’s movement experiences (e.g., rings, balls, mat, stilts, and ropes). Recess time in these ECE institutions ranged from 30 min to 45 min and took place in an outdoor environment and was a space to play freely without any educational purpose.

### 2.4. Classroom and Playground Densities

We measured the classroom and playground area (m^2^) in each school. In the case of playground area, we incorporated maps from Google Earth into geographic information system software (ArcGIS 10.2). We then calculated classroom and playground densities (preschoolers/m^2^), as described elsewhere [31].

### 2.5. Data Analysis

Analyses were undertaken using SPSS software, version 26.0, and the level of significance was set at *p* < 0.05. Descriptive statistics were used to summarise study variables and continuous data as mean and standard deviation. For the statistical analyses, we used one-way analysis of variance using the Bonferroni correction to estimate the effect of gender on MVPA. Mixed-effect regression analysis (using SPSS MIXED) evaluated the relationship between each independent variable with light PA, MVPA and SB, clustering participants within class groups (*n* = 10) and school groups (*n* = 6).

## 3. Results

The mean time (min) spent in each PA category during school hours is shown by gender in Figure 1. Compared to girls, boys engaged in significantly (*p* < 0.05) more minutes of light PA (23 SD 6 min vs. 20.2 SD 5.4 min), moderate PA (16.3 SD 6 min vs. 12.2 SD 5 min), vigorous PA (11.3 SD 6 min vs. 7.2 SD 5 min), MVPA (30.1 SD 12 min vs. 20 SD 8.4 min), and TPA (49 SD 21.3 min vs. 37.2 SD 16 min), while girls were observed in significantly more minutes of sedentary behaviour (195.4 SD 22.2 min vs. 209 SD 21 min; *p* < 0.05).

Table 2 presents data for minutes of MVPA during school hours over a mean of four days with versus without a structured movement session. The average time spent in MVPA during school hours on days without a structured movement session was 21.7 min (SD 10.2), compared to 40.9 min (SD 16.5) on days with a structured movement session. Boys engaged in significantly more minutes of MVPA than girls, both on days with a structured movement session (47 SD 16.2 vs. 34 SD 14) and on days without (26.1 SD 10.4 min vs. 17 SD 7.8 min) (*p* < 0.01).

Table 3 shows the MVPA observed during recess and structured movement sessions, relative to that seen over total school hours on days with a structured movement session. For boys and girls, MVPA during recess represented 22 and 20%, respectively, of the total MVPA observed during school hours. For boys, MVPA during the structured movement session represented half (49%) of the total MVPA during school hours, compared to 57% for girls.

Mixed-model regressions showed significant associations between gender and PA variables (Table 4). Compared to girls, boys had higher levels of PA, across intensity categories, and lower levels of SB (*p* < 0.05). Furthermore, participants’ age was positively related to MVPA and vigorous PA, and negatively related to SB (*p* < 0.05). Mixed-model regressions indicated a positive relationship between playground density and vigorous PA (*p* = 0.012). Finally, another positive association was found between classroom density and SB (*p* < 0.001).

## 4. Discussion

According to international recommendations, preschoolers should have at least 180 min per day of PA at any intensity level (TPA), and at least 60 min of MVPA [4]. To our knowledge, there is no accelerometry-based evidence in our country about how much PA children get during school hours. We observed that children had mean 43.2 (SD 20) min of TPA throughout the school day, a modest contribution (24%) to the minimum recommendation of 3 h of daily PA at any intensity level. Furthermore, none of the children reached the recommended level of 60 min of MVPA per day at school, and only 32% reached 30 min during school hours. In a five-day period, boys and girls reached just 42% (mean 25 min/day) of the minimum daily MVPA during school hours. These low levels of school-day MVPA have also been observed in other studies. Segura-Martínez et al. (2020) found that children had an average of 20 min of MVPA during the school day [32], while Chen et al. (2020) observed a median 29 min [33]. For their part, Lu et al. (2019) reported that children spent 6.6% of their day (both in and outside of school) in MVPA, and just 28% met the PA recommendations (32% of boys and 22% of girls) [18]. In the same line, Torres-Luque et al. (2016) showed that 93% of Spanish children achieved the minimum 60 min MVPA, but only 11.2% of children met the US National Association of Sport and Physical Education recommendation of 120 min/day [26]. These authors analysed the levels of PA over 14 h per day but did not examine the role of school in achieving the observed levels.

Other authors have found more positive results. In Norway, Nilsen et al. (2019) found that boys and girls spent, on average, 12% and 10% of their preschool day in MVPA, respectively [7], and Andersen et al. (2017) found that children had an average of 58 min (SD 20) of MVPA per day [19]. In the same country, Kippe and Lagestad (2018) found that preschool children achieved 58.4 min of daily MVPA during their time at school, and 39.8% of the children reached the international health recommendations of 60 min MVPA daily on weekdays [11]. Elsewhere in Scandinavia, Soini et al. (2014) found in Finland that only 20% of preschoolers had at least 120 min TPA, whereas 46% of children had over 60 min [20].

In Spain, more than 30% of young children (3–6 years) are obese or overweight [34]. They also spend an average of 6 h in ECE institutions. Our negative results (TPA contributing only 24% to recommended levels, and MVPA only 10%—and less for girls on both counts) clearly illustrates that our preschoolers are not moving enough to benefit their quality of life. Even though children spend most of their day at school, our results show more than 70% of daily PA has to take place during family and/or extracurricular activities in order to meet international recommendations. At the same time, national regulations indicate that schools must become true spaces for promoting healthy behaviours [35]. In light of our results, it is clear that ECE institutions must review their role as promoters of PA [36]. The new information in our study suggests that the low status of structured curricular PA practices in the ECE curriculum could be behind the low PA levels in our country. We agree with Nilsen et al. (2019) that the preschool arena is an ideal setting for promoting PA, providing unique and equitable opportunities for structured PA in all children, irrespective of their or their parents’ behaviours, attitudes, resources, and socioeconomic background [7].

Exploring the features of personal and sociocultural factors during school hours can help identify some possible explanations for the PA observed. Table 4 shows significant associations between gender and PA levels. Boys were more active than girls (22.9 min vs. 20.2 min light PA, 30.2 min vs. 19.6 min MVPA, 49 min vs. 37.2 min TPA). Other authors have reported similarly low levels of schooltime MVPA in girls. Andersen et al. (2017) found that only 32% of girls, compared to 67% of boys, reached the recommended level of 60 min of MVPA per day [19]. Kippe and Lagestad (2018) likewise showed that boys were more active than girls, resulting in more boys than girls meeting the health recommendations for PA [11]. In the Spanish context, Segura-Martínez et al. (2020) found that after an intervention modifying the indoor environment in an ECE classroom, MVPA during free indoor play time increased for all children at week 1 and for children at week 6 post-intervention, but this increase did not translate to an increase in MVPA during recess or throughout the school day in general [32]. In Chile, Kain et al. (2018) found that young girls were significantly less active during the entire school day. Such results suggest that reaching the recommended levels of MVPA during school time requires further efforts to encourage less-active children—with a special eye on gender differences—to increase intensity when doing activities related to locomotion, such as jumping, leaping, hopping, and running during structured movement sessions [37]. It seems that in the case of girls, structured PA opportunities could motivate them to get moving. Pate et al. (2004) suggested that 3- to 5-year-old girls participate in significantly less PA than boys during preschool, and a recent systematic review determined that MVPA decline begins in early to mid-childhood for both girls and boys [8]. However, the rate of decline was slightly greater in girls [38]. Thus, evidence supports early efforts to promote PA among girls; structured movement sessions during school hours could provide girls with needed opportunities to enhance movement experiences in ECE institutions.

International guidance advocates for laying the foundation for a healthy lifestyle by ensuring that young children get at least 120 min of PA a day: 60 min through structured activities and 60 min through non-structured opportunities [39]. For this reason, we decided to analyse the contribution of structured movement sessions on MVPA during school hours. To our knowledge, there are not studies examining the contribution of structured movement-based PA on children´s MVPA levels on days with and days without PE. When we calculated the mean minutes of MVPA on the day when there was a structured movement session, children spent twice as much time in MVPA (Table 2). This amount of time represents 22% of the recommended daily MVPA compared with the only 12% on school days without structured movement sessions. In other words, children spent almost twice as much time doing MVPA on days when they had structured PA opportunities. This difference was similar in boys and girls. This result is in keeping with the study by Kippe and Lagestad (2018), who found that PA during school hours was the main contributor to preschool children’s PA level on weekdays [11]. These authors found that the time children spent at preschool comprised 48.8% of the children’s total MVPA. According to the 180 min TPA recommended, during ECE school hours, boys and girls had 47 min and 34 min, respectively, of MVPA, or 26 and 19% of the time children spent at school. Furthermore, while girls showed significantly lower levels of PA across intensity categories than boys on school days both with and without structured movement sessions, these sessions generated a marked and positive increase in MVPA: the percentage of MVPA during school hours, relative to the international benchmarks (180 min TPA), increased from 9 to 19% on the day with a structured movement session.

Furthermore, the contribution of the structured movement session was more than double that seen during recess (Table 3). For boys and girls, MVPA accounted for 22% during recess and 20%, respectively, of the total MVPA observed during school hours. For boys, MVPA during the structured movement session represented half (49%) of the total MVPA during school hours, compared to 57% for girls. Our results suggest that structured PA is an important strategy for offering movement experiences to young children. Following a different structured PA opportunity (classroom-based PA breaks), Wadsworth et al. (2012) found that these accounted for 69% of the children’s daily MVPA in one childcare center and 90% in another [24]. In secondary education, Frömel et al. (2016) found that PE sessions had a central role in school-based PA [15]. In the elementary school context, Tyler et al. (2020) found that on days when only PE was provided, boys had an additional 5.2 min of MVPA and girls, 3.1 min [40]. In primary education in Switzerland, Meyer et al. (2013) found that children were significantly more active on days with PE compared with days without PE [41]. In primary education in Spain, Martínez et al. (2012) found that MVPA carried out during the PE session represented a high percentage of weekly healthy PA [42]. Furthermore, the children who had more minutes of vigorous PA within the PE session also had more minutes of vigorous PA throughout the week. We observed that the contribution of the structured movement sessions was more positive in the case of young girls, nearly tripling the amount of MVPA compared with days without this kind of session. In other words, our results show empirically that structured PA in the ECE context could reduce the gender gap between girls and boys in terms of PA promotion at a young age, and more PA opportunities (and in particular structured PA) could help to reduce the gap in achieving international recommendations.

One possible explanation for the difference in PA levels between recess and structured movement sessions is related to the nature of free play during recess. Verbestel et al. (2011) found three notable peaks in PA during preschool hours (morning recess, lunch break, and afternoon recess) [13], and Pate et al. (2004) suggested that the time that children are allowed to play freely in settings that are conducive to PA probably exerts a strong influence on PA in preschools [8]. In fact, these authors argue that one potentially effective strategy for providing children with adequate PA is to allow them ample time in recess and other free-play settings. However, the literature reports that recess time in young children is actually associated with low levels of moderate to vigorous activity [17,43]. For example, Lu et al. (2019) found that recess and afternoon only accounted for 20% of children’s total daily MVPA [18]. Those results are consistent with ours, which showed that recess time may not be as effective for promoting PA in preschoolers as structured PA.

The results of the regressions suggest some possible explanations for the MVPA observed during school hours: a high density of young children on the playground was associated with high levels of vigorous PA, whereas, in the classroom, high density was associated with more sedentary behaviour. Regarding the higher density on the playground, the literature suggests that increased opportunities for interactive play could impact PA. For instance, Terrón-Pérez et al. (2019) found that 71% of the intervals observed during recess time were categorised as ‘interaction’, suggesting that the playground is a space for social relations, and that this was a predictor of higher MVPA levels [17]. In the same line, Miranda et al. (2017) found that both boys and girls who engaged in more group play during recess time showed greater involvement than those who engaged more in solitary play [44]. However, in our sample, recess time did not induce higher PA levels; on the contrary, the contribution of recess time to the total PA during school hours was notably less than that of structured movement sessions. Therefore, even though our results suggest that social circumstances during recess time (higher density in the playground) could predict vigorous PA in preschoolers during school hours, our data also showed that on school days with structured movement sessions, children had significantly higher PA levels than on days when opportunities for PA were limited to unstructured recess. Thus, we concur with Lahuerta-Contell (2021) that increasing the number of school days when preschoolers can participate in structured movement sessions is warranted [29].

Regarding the number of children per classroom, we found that a high density of students was associated with higher levels of SB during school hours. This result corroborates findings in the Danish context [23], which showed a positive association between MVPA and the indoor area per child. Likewise, recent research has shown that certain modifications in indoor spaces in the ECE context can have an impact on PA [32,45], so larger classrooms and/or lower student/teacher ratios could also increase opportunities for movement.

Age was a positive predictor of vigorous PA in our sample (Table 4). Specifically, older preschoolers performed 2 more minutes of vigorous PA than younger ones. Similarly, a recent study in Belgium found that older preschoolers were more likely to meet all guidelines on weekdays than the younger children [46]. For the authors, the benefits of a structured school day increase as preschoolers get older. In Norway, Nilsen et al. (2019) also found that the preschool arena affects children’s MVPA increasingly with age [7]. Our results add to the evidence base in this regard, giving more weight to the contention that older children benefit more from structured ECE environments than younger children in terms of high-intensity activities, and they have more capacity to move with higher intensities.

### Strengths and Limitations

This is the first study in our country that objectively measure PA levels in young children attending ECE institutions. Our results show empirically that structured PA in the ECE context could reduce the gender gap between girls and boys in terms of PA promotion at a young age, and more PA opportunities (in particular structured PA) could help to reduce the gap in achieving international recommendations. However, the study does have some limitations. First of all, despite the fact that PA levels were measured through accelerometry, this method alone was insufficient for determining at the same time, the contextual circumstances of PA in young children during the two specific opportunities to be physically active throughout the day. Further studies should use systematic observation to study contextual variables such as the presence of others, location, availability of equipment, and gender-based group interactions in the TPA and MVPA during school hours. Secondly, because this study was performed only during school-hours we did not take into account how families could counteract the very little TPA for young children attending ECE institutions. Therefore, further research should explore how structured movement sessions and recess time in the ECE context could act as a predictor of home time and/or extra-curricular PA. We hope our research is shared among ECE practitioners, preservice practitioners and students; the ECE community should take into account that ECE institutions must review their role as promoters of PA.

## 5. Conclusions

Our main findings were that: (a) preschoolers engaged in very little TPA and MVPA during school hours in ECE institutions, falling far short of compliance with international and national recommendations on PA; (b) children showed significantly higher MVPA levels on days with versus without structured movement sessions; (c) the contribution of the structured movement sessions to total MVPA was significantly higher than that of recess time; (d) boys were significantly more active than girls in all PA intensity categories; (e) younger preschoolers were less active than older preschoolers; (f) on the playground, a high density of young children was associated with higher levels of vigorous PA; (g) in the classroom, a high density of students was associated with more sedentary behaviour.

Structured movement sessions accounted for a significant proportion of school-time PA; however, we cannot expect this kind of structured PA opportunity to provide all the daily PA that young children need. Although ECE institutions are important environments for establishing good PA behaviours, the bulk of the responsibility for promoting healthy behaviours currently falls on families. However, broader social trends, for example, impacting the work–life balance and the social support available to families, may be major barriers for promoting leisure-time PA in young children [47]. Like other research teams [32,48,49], we believe that schools should complement structured movement opportunities with models that increase opportunities for PA throughout the school day. ECE institutions should assume greater responsibility for offering children adequate movement experiences. Furthermore, given the major challenges to leisure-time PA that young children and adults are facing in the new COVID-19 context, ECE institutions have a responsibility to minimise, insofar as they can, these impacts on PA levels.

## Figures and Tables

**Figure 1 children-08-00082-f001:**
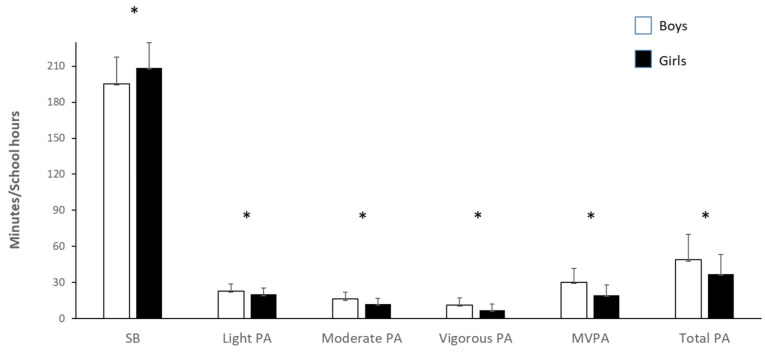
Categories of physical activity during school hours based on gender. SB, sedentary behaviour; LPA, light physical activity; MPA, moderate physical activity; VPA, vigorous physical activity; MVPA, moderate to vigorous physical activity; Total PA: light, moderate, or vigorous PA. * *p* < 0.05 between boys and girls.

**Table 1 children-08-00082-t001:** Characteristics of preschool children (N = 116).

Variable	Measure
Age in years, mean (SD)	4.3 (0.5)
Gender, *n* (%)	
Boys	51.3%
Girls	48.7%
Height (cm), mean (SD)	102.4 (6.3)
Weight (kg), mean (SD)	18 (3.5)

SD: standard deviation.

**Table 2 children-08-00082-t002:** Mean (standard deviation) minutes of MVPA on school days with (*n* = 1) versus without (*n* = 4) structured movement sessions.

	Gender		
	Boys	Girls	Total
Day with structured movement session	47 (16.2)	34 (14) *	40.9 (16.5) **
Days without structured movement session	26.1 (10.4)	17 (7.8) *	21. 7 (10.2)

* *p* < 0.01 between boys and girls; ** *p* < 0.01 between with and without structured movement session.

**Table 3 children-08-00082-t003:** MVPA during total school day, recess time, and structured movement sessions, and relative contribution of the MVPA carried out during these designated time periods to total MVPA during school hours, by gender.

	Mean min (SD) MVPA			Relative Contribution to Total MVPA during School Hours during Designated Periods, % (SD)	
	Total School Day	Recess Time	Structured Movement Session	Recess Time	Structured Movement Session
Boys	47 (16) *	9 (6.4)	23 (9) *	22% (8)	49% (18.1) **
Girls	34 (14)	8 (6)	18 (8)	20% (8)	57% (18.3)

* *p* < 0.01 between boys and girls; ** *p* = 0.05 between boys and girls.

**Table 4 children-08-00082-t004:** Mixed-model regression results for relationship between independent variables and physical activity levels during school hours in preschoolers, according to accelerometry.

	Light PA ^a^			MVPA ^a^			Moderate PA ^a^			Vigorous PA ^a^			Sedentary Behaviour ^a^		
	*β*	*t*	*p*	*β*	*t*	*p*	*β*	*t*	*p*	*β*	*t*	*p*	*β*	*t*	*p*
Boys	2.6	2.4	0.01	10.7	5.9	<0.01	4.8	4.4	<0.01	5.1	4.7	<0.01	−15.4	−5.5	<0.01
Girls	1			1			1			1			1		
Age	0	0.8	0.39	0.01	2.1	0.03	0	0.2	0.81	0	2	0.04	−0.0	−2.7	0.01
Classroom density	−0.8	−0.8	0.42	1.1	0.6	0.52	1.3	1.2	0.21	1	1	0.3	13.9	3.6	<0.01
Playground density	−0.3	−1.4	0.16	0.7	1.6	0.1	0	0	0.99	0.5	2.5	0.01	0.1	0	0.96

^a^ Participant clustering in class groups and school groups was adjusted for as a random effect in the models. MVPA: moderate to vigorous physical activity; PA: physical activity.

## Data Availability

The data presented in this study are available on request from the corresponding author. The data are not publicly available due to privacy or ethical restrictions.

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
