# Peer review of "The Role of Preschool Hours in Achieving Physical Activity Recommendations for Preschoolers"

_children, 2021, doi:10.3390/children8020082_

Round 1
Reviewer 1 Report
This is very interesting research on children's behavior at school and about the school's role in promoting health and exercise in children, which is a primary matter in physical literacy.
The manuscript is well written, and the discussion very thorough and extended. In my opinion, the authors have to address only a few issues and comments.
- Abstract: the study is a descriptive report of children's PA behaviors. However, the abstract might synthesize your conclusions as in the manuscript, differently than just reporting the results.
- Consider being consistent in the structure: the study pointed to three aims (a to c). In the results and conclusion, a different structure was used (a to g). Readers might benefit from a more consistent structure.
- Methods: please describe more thoroughly the structured movement sessions? Do they had been planned before or just led by the teachers? What kind of activities did the children attend? How do they have been managed? Do they were only allowed to play by the teachers guiding some games or activities were more structured? What sort of teaching approaches has been used?
- Does the children's recess time has been observed? I think that this is a pivotal point to consider and discuss: a higher PA when children are guided in exercising/playing is somehow reasonably to be expected compared to when only recess time is allowed (because of leading exercise, of course). However, if the authors observed activities and recess time, more details would be of great interest.
- Table 3. Symbols (* and cross) generate some confusion because they were previously used in Table 2 differently. I suggest being more detailed and specifying comparisons in the footnotes, and possibly using * and **.
- Table 4. the table is difficult to be read because of the format. Please keep the data of each variable in a single line, by reducing decimals. Delete "Gender", perhaps "Variable" in the column titles.
General comments: - Guidelines from the literature generally suggest preschoolers to spend a certain amount of PA per day/week. This study has been designed to investigate only the PA spent at school and give interesting information. However, the authors did not report the estimated minutes of PA that children usually performed daily or weekly, to relate to the PA measured at school. It has not been reported, surveyed, considered because of some reasons, or is this a limitation of the study? May the results be somehow affected by the children's "predisposition" to move? In your opinion, may an increase of PA with structured sessions be affected by a "low" or "high" tendency of children to move? In other words: shy, overweight, quiet children possibly exercise during recess time differently than generally active and exuberant children. However, guided activities aim to induce everyone to exercise. What about considering as a variable also a measured level of daily PA of children?
- Finally, does the structured activities have been led by specialists or generalists? Specialists have been found to lead the PA structured activities better than generalists, increasing PA levels, enjoyment, and satisfaction to exercise. This is a further argument to be considered and perhaps discussed.
Author Response
- Abstract: the study is a descriptive report of children's PA behaviors. However, the abstract might synthesize your conclusions as in the manuscript, differently than just reporting the results. Consider being consistent in the structure: the study pointed to three aims (a to c). In the results and conclusion, a different structure was used (a to g). Readers might benefit from a more consistent structure.
Answer:
We have modified the abstract accordingly.
New abstract:
Abstract: Research on physical activity (PA) in different educational settings could elucidate which inter-ventions promote a healthy school lifestyle in early childhood education (ECE). The aims of this study were: (a) to analyse the PA levels of preschoolers during school hours, as well as the rate of compliance with specific recommendations on total PA (TPA) and moderate-vigorous PA (MVPA); (b) to examine the role of structured movement sessions and recess time in the MVPA levels during school hours; and (c) to evaluate the sociodemographic correlates of preschoolers and the school environment on MVPA behaviour during school hours. PA was evaluated with Actigraph accelerometers. Our main findings were that: (a) preschoolers engaged in very little TPA and MVPA during school hours; (b) children showed significantly higher MVPA levels on days with versus without structured movement sessions, and the contribution of the structured sessions to MVPA was significantly higher than that of recess time; and (c) gender and age were associated with PA, and a high density of young children on the playground was associated with high levels of vigorous PA, whereas in the classroom, high density was associated with more sedentary behaviour. Structured PA could reduce the gap in achieving international recommendations.
- Methods: please describe more thoroughly the structured movement sessions? Do they had been planned before or just led by the teachers? What kind of activities did the children attend? How do they have been managed? Do they were only allowed to play by the teachers guiding some games or activities were more structured? What sort of teaching approaches has been used?
Answer:
Following the reviewer advice we have given more information about the structured movement sessions.
New paragraph:
The structured movement sessions were structured (i.e. following guidelines to achieve educational goals) and were part of the curriculum of the preschools participating in the research, that is, they had a justification, objectives, procedure, and evaluation. The teachers structured the sessions according to the guidelines of their own curriculum. The sessions took place once a week, were carried out during the regularly scheduled time period, were held in an indoor classroom, lasted approximately 45 min, and 14 to 20 children participated. The same number of children participated in the session as during the whole school day. On the day of the measurement, the teachers were instructed to conduct the session the way they usually do. The indoor classroom was equipped with typical early childcare equipment in order to stimulate children’s movement experiences (e.g. rings, balls, mat, stilts, and ropes). Recess time in these ECE institutions ranged from 30 min to 45 min and took place in an outdoor environment and was a space to play freely without any educational purpose.
- Does the children's recess time has been observed? I think that this is a pivotal point to consider and discuss: a higher PA when children are guided in exercising/playing is somehow reasonably to be expected compared to when only recess time is allowed (because of leading exercise, of course). However, if the authors observed activities and recess time, more details would be of great interest.
Answer:
Children were no observed during recess time; instead we have measured PA levels using accelerometers. Furthermore, because not all structured sessions were observed, we found it necessary to focus our analysis only on PA levels measured through accelerometry. We agree with the reviewer that accelerometry alone is insufficient for determining the contextual circumstances of PA in young children, and further studies should use systematic observation to study contextual variables such as the presence of others, location, availability of equipment, and gender-based group interactions. We have included this as a limitations of the study. We have included a new paragraph.
New paragraph:
Strengths and Limitations
This is the first study in our country that objectively measure PA levels in young children attending ECE institutions. Our results show empirically that structured PA in the ECE context could reduce the gender gap between girls and boys in terms of PA promotion at a young age, and more PA opportunities (and in particular structured PA) could help to reduce the gap in achieving international recommendations. However, the study does have some limitations. First of all, despite the fact that PA levels were measured through accelerometry, this method alone was insufficient for determining at the same time, the contextual circumstances of PA in young children during the two specific opportunities to be physically active throughout the day. Further studies should use systematic observation to study contextual variables such as the presence of others, location, availability of equipment, and gender-based group interactions in the TPA and MVPA during school hours. Secondly, because this study was performed only during school-hours we did not take into account how families could counteract the very little TPA for young children attending ECE institutions. Therefore, further research should explore how structured movement sessions and recess time in the ECE context could act as a predictor of home time and/or extra-curricular PA. We hope our research is shared among ECE practitioners, preservice practitioners and students; the ECE community should take into account that ECE institutions must review their role as promoters of PA.
- Table 3. Symbols (* and cross) generate some confusion because they were previously used in Table 2 differently. I suggest being more detailed and specifying comparisons in the footnotes, and possibly using * and **.
Answer:
Following reviewer advice we have modified accordingly.
New footnotes:
Table 2. *p<0.01 between boys and girls; **p<0.01 between with and without structured movement session.
Table 3. *p<0.01 between boys and girls; **p=0.05 between boys and girls.
- Table 4. the table is difficult to be read because of the format. Please keep the data of each variable in a single line, by reducing decimals. Delete "Gender", perhaps "Variable" in the column titles.
Answer:
We have modified the table accordingly.
- General comments:
Guidelines from the literature generally suggest preschoolers to spend a certain amount of PA per day/week. This study has been designed to investigate only the PA spent at school and give interesting information. However, the authors did not report the estimated minutes of PA that children usually performed daily or weekly, to relate to the PA measured at school. It has not been reported, surveyed, considered because of some reasons, or is this a limitation of the study?
Answer:
We have only analysed the school hours because research has been focused the attention during all day but little is known about the school hours, and in particular the impact of the structured and non-structured activities in PA. This is the first study in our country that objectively measure PA levels in young children attending ECE institutions. This was the main reason why we did not report non-school hours. Following reviewer recommendation, we have included a new paragraph as a Limitations of the study.
New paragraph:
Strengths and Limitations
This is the first study in our country that objectively measure PA levels in young children attending ECE institutions. Our results show empirically that structured PA in the ECE context could reduce the gender gap between girls and boys in terms of PA promotion at a young age, and more PA opportunities (and in particular structured PA) could help to reduce the gap in achieving international recommendations. However, the study does have some limitations. First of all, despite the fact that PA levels were measured through accelerometry, this method alone was insufficient for determining at the same time, the contextual circumstances of PA in young children during the two specific opportunities to be physically active throughout the day. Further studies should use systematic observation to study contextual variables such as the presence of others, location, availability of equipment, and gender-based group interactions in the TPA and MVPA during school hours. Secondly, because this study was performed only during school-hours we did not take into account how families could counteract the very little TPA for young children attending ECE institutions. Therefore, further research should explore how structured movement sessions and recess time in the ECE context could act as a predictor of home time and/or extra-curricular PA. We hope our research is shared among ECE practitioners, preservice practitioners and students; the ECE community should take into account that ECE institutions must review their role as promoters of PA.
- May the results be somehow affected by the children's "predisposition" to move? In your opinion, may an increase of PA with structured sessions be affected by a "low" or "high" tendency of children to move? In other words: shy, overweight, quiet children possibly exercise during recess time differently than generally active and exuberant children. However, guided activities aim to induce everyone to exercise. What about considering as a variable also a measured level of daily PA of children?
Answer:
Thanks to the reviewers for this comment. According to our results, the contribution of the structured movement session was more than double that seen during recess, suggesting that structured PA is an important strategy for offering movement experiences to young children. However, as the reviewer suggest, it could be possible that some contextual and/or psychosocial factors could impact PA. As we have indicated previously, further studies should use systematic observation to study contextual variables such as the presence of others, location, availability of equipment, and gender-based group interactions. By the other hand, according to the relationship between BMI/waist circumference we did not find these as predictor of TPA/MVPA. For instance, according to our results, in our sample differences between overweight vs nor-overweight children and PA were not seen.
- Finally, does the structured activities have been led by specialists or generalists? Specialists have been found to lead the PA structured activities better than generalists, increasing PA levels, enjoyment, and satisfaction to exercise. This is a further argument to be considered and perhaps discussed.
Answer:
The structured movement sessions were coordinated by an ECE teacher. According to the standardized curriculum, the Spanish ECE curriculum emphasises knowledge of one's own body, of others, and of one's possibilities of action. In our country, ECE teachers have traditionally been trained as generalists. However, there is now a growing awareness of the potential importance of pedagogical content knowledge of PA opportunities in the ECE curriculum. In our country, the structured PA are led by the same teacher who stay in the traditional classroom with the children. Following reviewer comment we have included a new paragraph about the structured movement sessions.
New paragraph:
The structured movement sessions were structured (i.e. following guidelines to achieve educational goals) and were part of the curriculum of the preschools participating in the research, that is, they had a justification, objectives, procedure, and evaluation. The teachers structured the sessions according to the guidelines of their own curriculum. The sessions took place once a week, were carried out during the regularly scheduled time period, were held in an indoor classroom, lasted approximately 45 min, and 14 to 20 children participated. The same number of children participated in the session as during the whole school day. On the day of the measurement, the teachers were instructed to conduct the session the way they usually do. The indoor classroom was equipped with typical early childcare equipment in order to stimulate children’s movement experiences (e.g. rings, balls, mat, stilts, and ropes). Recess time in these ECE institutions ranged from 30 min to 45 min and took place in an outdoor environment and was a space to play freely without any educational purpose.
Reviewer 2 Report
Method section: “PA levels were measured with Actigraph accelerometers (GT3X+Actigraph, Pen91 sacola, FL). Accelerometers are an objective measurement device and have been validated to assess PA in the preschool population” –need to cite the reference.
The number of boys and girls of total 116 sample size.
Exclusion and inclusion criteria
How the sample size were calculated?
Discussion section line 259 paragraph needs to be expanded rather than citing table 3.
Limitations of this study need to be addressed.
Author Response
- Method section: “PA levels were measured with Actigraph accelerometers (GT3X+Actigraph, Pensacola, FL). Accelerometers are an objective measurement device and have been validated to assess PA in the preschool population” –need to cite the reference.
Answer:
We have included the references.
New text:
Accelerometers are an objective measurement device and have been validated to assess PA in the preschool population (Cardon & De Bourdeaudhuij, 2008; Pate et al., 2004; Van Cauwenberghe et al., 2011).
The number of boys and girls of total 116 sample size.
Answer:
We have modified the paragraph.
New text:
We included 116 young children aged three to four years (mean age 4.3 (SD = 0.5); 59 girls) from 6 different public ECE institutions in the province of Valencia, Spain (Table 1).
- Exclusion and inclusion criteria
Answer:
We appreciate the reviewer’s comment about inclusion and exclusion criteria. We have now included information about this in two sections of the manuscript, as follows:
- 2.1. Design and participants section: “Inclusion criteria were: children aged 3-6 years and being able to walk without assistance.”
- 2.2. Objective physical activity measurement section: “Children (n = 20) who did not have 5 consecutive days registered were excluded (e.g., due to illness or lack of attendance at the school on the days when measurements were carried out) from the final analyses.”
- How the sample size were calculated?
Answer:
The sample size was not established prior to the study, but was broadly guided by what is typical in other studies in the literature on physical activity among preschoolers.
- Discussion section line 259 paragraph needs to be expanded rather than citing table 3.
Answer:
We appreciate this reviewer’s comment. We have modified the paragraph.
New text:
Furthermore, the contribution of the structured movement session was more than double that seen during recess (table 3). For boys and girls, MVPA accounted for 22% during recess and 20%, respectively, of the total MVPA observed during school hours. For boys, MVPA during the structured movement session represented half (49%) of the total MVPA during school hours, compared to 57% for girls. Our results suggest that structured PA is an important strategy for offering movement experiences to young children.
- Limitations of this study need to be addressed.
Answer:
Following reviewer advice we have included a Strengths and Limitations section.
New paragraph:
Strengths and Limitations
This is the first study in our country that objectively measure PA levels in young children attending ECE institutions. Our results show empirically that structured PA in the ECE context could reduce the gender gap between girls and boys in terms of PA promotion at a young age, and more PA opportunities (and in particular structured PA) could help to reduce the gap in achieving international recommendations. However, the study does have some limitations. First of all, despite the fact that PA levels were measured through accelerometry, this method alone was insufficient for determining at the same time, the contextual circumstances of PA in young children during the two specific opportunities to be physically active throughout the day. Further studies should use systematic observation to study contextual variables such as the presence of others, location, availability of equipment, and gender-based group interactions in the TPA and MVPA during school hours. Secondly, because this study was performed only during school-hours we did not take into account how families could counteract the very little TPA for young children attending ECE institutions. Therefore, further research should explore how structured movement sessions and recess time in the ECE context could act as a predictor of home time and/or extra-curricular PA. We hope our research is shared among ECE practitioners, preservice practitioners and students; the ECE community should take into account that ECE institutions must review their role as promoters of PA.
Reviewer 3 Report
Many thanks for this read, it was an interesting topic that may become more pertinent post COVID-19 where activity may drop due to a lack of outside time and limited contact with friends.
I have a few notes that I have made which you may have addressed and I missed, however:
- Was there a reason for the choice of Actigraph as the manufacturer, are they superior in any way to alternatives?
- Did you consider other activity/intensity measurements such as heart rate monitoring or GPS to track distances?
- A further in depth discussion into past activity research before the advent of smartphones, technology and other potential reasons for a downturn in child activity. Does past research show greater levels of activity from the 1970s, 80s, 90s etc.?
Author Response
- Was there a reason for the choice of Actigraph as the manufacturer, are they superior in any way to alternatives?
Answer:
The Actigraph is one of the most commonly used accelerometer in physical activity research involving children and adolescent (Trost et al., 2011). Moreover, the Actigraph accelerometer is one of the few whose validity and reliability has been specifically analysed in preschool (Cardon et al., 2008; Pate et al., 2004; Van Cauwenberghe et al., 2011). Likewise, the cut-off points in this population have been specifically analysed using the Actigraph (Van Cauwenberghe et al., 2011). That is why we decided to use it in the present study.
References:
Cardon LR, De Bourdeaudhuij MM (2008) Are preschool children active enough? Objectively measured physical activity levels. Res Q Exerc Sport 79:326–332.
Pate RR, Pfeiffer KA, Trost SG, Ziegler P, Dowda M (2004) Physical activity among children attending preschools. Pediatrics 114:1258–1263.
Trost SG, Loprinzi PD, Moore R, Pfeiffer KA (2011) Comparison of accelerometer cut points for predicting activity intensity in youth. Med Sci Sports Exerc 43:1360–1368.
Van Cauwenberghe, E., Labarque, V., Trost, S. G., De Bourdeaudhuij, I., & Cardon, G. (2011). Calibration and comparison of accelerometer cut points in preschool children. International Journal of Pediatric Obesity, 6(2Part2), e582-e589.
- Did you consider other activity/intensity measurements such as heart rate monitoring or GPS to track distances?
Answer:
We did not consider other physical activity measurements. In our opinion, considering our study purposes, accelerometry was the most appropriate method to evaluate physical activity due to the necessity to know the total number of minutes spent in each type of physical activity (i.e., light, moderate and vigorous activity) as well as sedentary time.
- A further in depth discussion into past activity research before the advent of smartphones, technology and other potential reasons for a downturn in child activity. Does past research show greater levels of activity from the 1970s, 80s, 90s etc.?
Answer:
We are very sorry, but there is a lack of studies from the 1970s, 80s or 90s that have specifically analysed the levels of physical activity during school hours in preschoolers. Furthermore, the measurement of physical activity in those decades was based mainly on the use of subjective measures (e.g., questionnaire) and not on objective measures such as accelerometers. This would make data comparison more difficult.